# Photodynamic Therapy for Glioblastoma: Potential Application of TiO_2_ and ZnO Nanoparticles as Photosensitizers

**DOI:** 10.3390/pharmaceutics17091132

**Published:** 2025-08-29

**Authors:** Emma Ortiz-Islas, María Elena Manríquez-Ramírez, Pedro Montes, Citlali Ekaterina Rodríguez-Pérez, Elizabeth Ruiz-Sanchez, Karla Carvajal-Aguilera, Victoria Campos-Peña

**Affiliations:** 1Laboratorio de Neurofarmacología y Nanotecnología Molecular, Instituto Nacional de Neurología y Neurocirugía, Manuel Velasco Suárez, Mexico City 14269, Mexico; emma.ortiz@innn.edu.mx (E.O.-I.); crodriguez@innn.edu.mx (C.E.R.-P.); 2Laboratorio de Investigación en Nanomateriales y Energías Limpias, ESIQIE-IPN, Instituto Politécnico Nacional, Lindavista 07700, Gustavo A. Madero CDMX, Mexico City 14269, Mexico; marymanriquez@yahoo.com.mx; 3Laboratorio de Neuroinmunoendocrinología, Instituto Nacional de Neurología y Neurocirugía, Manuel Velasco Suárez, Mexico City 14269, Mexico; pedrovolvox@gmail.com; 4Laboratorio de Neuroquímica, Instituto Nacional de Neurología y Neurocirugía, Manuel Velasco Suárez, Mexico City 14269, Mexico; elizabeth.ruiz@innn.edu.mx; 5Laboratorio de Nutrición Experimental, Instituto Nacional de Pediatria, Mexico City 04530, Mexico; karla_ca@yahoo.com; 6Laboratorio Experimental de Enfermedades Neurodegenerativas, Instituto Nacional de Neurología y Neurocirugía, Manuel Velasco Suárez, Mexico City 14269, Mexico

**Keywords:** glioblastoma, photodynamic therapy, photosensitizers, nanoparticles, titanium dioxide, zinc oxide

## Abstract

Despite aggressive current therapies against glioblastoma (GB), residual tumor cells may remain at the edge of the surgical cavity after resection. These cells can rapidly proliferate, giving rise to tumor recurrence in more aggressive and drug-resistant forms. As photodynamic therapy (PDT) has advanced, it has emerged as an option to treat this brain tumor. The oncological basis of PDT involves the selective accumulation of a photosensitizer (PS) in the tumor, followed by its activation with electromagnetic radiation to generate reactive oxygen species (ROS), which induce tumor cell death. Given that first- and second-generation PSs present significant limitations, including poor tumor selectivity, suboptimal biodistribution, limited absorption within the therapeutic window, and slow systemic clearance, research has progressed toward the development of third-generation PSs based on nanotechnology to optimize their therapeutic properties. This review addresses the types of tumor cell death induced by PDT, as well as the advancements of PS design, focusing on titanium dioxide (TiO_2_) and zinc oxide (ZnO) nanoparticles. These nanomaterials can be designed as carriers, encapsulating or conjugating conventional PSs, or act as PSs themselves, due to their favorable biocompatibility and intrinsic photoreactivity. Additionally, they can be functionalized with targeting ligands to achieve tumor-specific delivery, enhancing therapeutic selectivity while minimizing toxicity to healthy tissue. Overall, these nanotechnology-based PSs represent a versatile and promising therapeutic paradigm that warrants further investigation through basic research, supporting the development and potential clinical translation of a more precise and effective PDT-based intervention for glioblastoma, initially aimed at eliminating intra-surgical post-resection residual tumor cells.

## 1. Introduction

Gliomas are the most common primary tumors of the central nervous system (CNS), with glioblastoma (GB) (WHO grade IV) accounting for approximately 50% of all malignant CNS tumors [1]. GB is considered the most lethal CNS tumor and occurs primarily in adults over the age of 65. With a global annual incidence of 3.22 per 100,000, it accounts for 54.7% of all gliomas and 16% of all primary malignant neoplasms of the brain and CNS. Although it is a rare tumor compared to other cancers, it is one of the top ten causes of death, accounting for 4% of all tumor deaths. The “Stupp protocol” is the current standard of care for newly diagnosed GB. It consists of gross surgical resection of the tumor (if feasible) followed by radiotherapy (RT) and chemotherapy with concurrent temozolomide for 6 weeks and adjuvant for 6 months [2]. However, complete surgical resection is nearly impossible due to its high cellular heterogeneity and intrinsic ability to invade adjacent healthy tissue. Tumor cells remaining at the edge of the cavity after surgical removal of the tumor mass are resistant to standard therapies, as the majority of GBs quickly re-emerge 2–3 cm from the original tumor site and give rise to more aggressive and resistant forms of the disease [3]. The prognosis for patients with high-grade gliomas is poor, with a median survival of approximately 9 to 12 months, despite recent advances in surgical techniques and postoperative management [1].

Chemotherapy is known to be associated with systemic side effects. Radiotherapy, on the other hand, is limited by the combined radiation dose. As a result, research has focused on developing alternative therapeutic approaches that are more specific and selective, safe, effective, and economical. One such approach is photodynamic therapy (PDT), which is a modern, non-invasive, and rapid method for the diagnosis and treatment of various diseases, including cancer, bacterial infections, psoriasis, and atherosclerosis, among others [4,5,6,7]. PDT in particular is considered an ideal cancer treatment because it damages cancer cells by forming reactive oxygen species (ROS) when a photosensitizing agent is irradiated with light [6,7,8]. Compared to traditional cancer treatment modalities (surgery, chemotherapy, radiation therapy, and immunotherapy), PDT has the advantages of high safety, reproducibility, low long-term morbidity, and high quality of life for patients, and selectivity for cancer cells, damaging only cancer cells and sparing surrounding healthy cells and tissues.

A crucial determinant of successful PDT is the selection of PSs with optimal physicochemical and photobiological properties, specifically the ability to be activated by light within the optimal therapeutic window (600–800 nm), where tissue penetration into the tumor is maximized and phototoxicity to surrounding healthy tissue is minimized. In the neuro-oncological setting, the blood–brain barrier, tumor heterogeneity, and other factors present additional challenges for conventional PSs within precision medicine frameworks. These difficulties have prompted exploration of alternative strategies. To specifically target the site of the tumor, a light-generating agent would need to be placed. This approach is still at an early stage, but has been explored in other cancer types through the use of self-luminescent nanosystems [9]; consequently, current advances have focused on the perioperative induction of death in persistent tumor cells by PDT following glioblastoma surgical resection [10]. This work addresses PDT, including the types of cell death it can induce. Particular emphasis is placed on the evolution of PS optimization, highlighting titanium dioxide (TiO_2_) and zinc oxide (ZnO) nanoparticles as third-generation nanotherapeutic PSs for glioblastoma, given their potential to enhance bioavailability, improve tumor targeting, and increase intratumoral cytotoxicity.

## 2. Photodynamic Therapy (PDT)

Basically, PDT depends on three main components: (1) radiation (600–1000 nm), (2) a photosensitizer (PS), and (3) oxygen in its molecular state (O_2_) [11,12,13,14]. The PDT procedure is initiated by the local or intravenous administration of a PS to the intended site of treatment. Subsequently, radiation at an appropriate wavelength is applied in order to trigger the desired photosensitization reaction. This causes the PS_0_ transition from its basal (low energy) state to an unstable, excited singlet state (PS_1_) with higher energy and a shorter lifetime. PS_1_ can return to its basal state by releasing excess energy, often emitted as fluorescence. Alternatively, it can undergo intersystem crossover, transitioning to a higher-energy, longer-lived triplet excited state (PS_3_) [15,16]. PS_3_ is also unstable and can return to its basal state by releasing energy in the form of phosphorescence. Alternatively, PS_3_ can transfer this energy to other molecules, such as O_2_, creating ROS through two different pathways, as shown in Figure 1.

### 2.1. Type I Reactions in PDT

This reaction occurs when the PS_3_ state interacts with a substrate (ST) through electron or proton transfer, forming cationic or anionic radicals [17]. These radicals can rapidly react with the O_2_ present in the cell to produce ROS, such as superoxide anions (O_2_^−^), hydroxyl radicals (OH^−^), and hydrogen peroxide (H_2_O_2_), as shown on Equations (1)–(4). It has been reported that OH^−^ is a highly reactive species damaging proteins and cell membrane structures through lipid peroxidation [18]. In contrast, H_2_O_2_ does not cause significant oxidative damage but has a long half-life and can undergo the Fenton reaction with O_2_^−^ to produce OH^−^ [19].PS_0_ + λν → PS_1_→ PS_3_(1)PS_3_ + ST → PS_3−_ + ST^+^ (radicals)(2)PS_3−_ + O_2_ → PS + O^2−^(3)ST^+^ + H_2_O → OH^−^ + H_2_O_2_(4)

### 2.2. Type II Reactions in PDT

In type II reactions, PS_3_ reacts with O_2_ through energy transfer to form singlet oxygen (^1^O_2_), a highly reactive species (see Equations (5) and (6)) [20,21]. Singlet oxygen is more stable and highly oxidant with substrates such as nucleic acids and proteins, leading to cell destruction. Therefore, singlet oxygen is more desirable than radicals due to its longer lifetime.PS_3_ + O_2_ → PS_0_ → ^1^O_2_
(5)^1^O_2_ + ST → ST (oxidized)(6)

Type I and type II reactions can coexist, and their contribution to the final PDT efficiency depends on factors such as the type of PS, oxygen O_2_ concentration, substrate concentration, PS–substrate binding capacity, and cellular environment [22]. The type II reaction is considered the most prominent mechanism, and most PSs follow this type of reaction.

Some reports of a type III reaction have been found, and it has been suggested that it occurs alongside types I and II [17]. This reaction reportedly takes place between PS_3_ and the free radicals present in the system. It is also referred to as a triplet–doublet process or a modified type I reaction. Very little is known about these reactions. However, type I and type II reactions are considered the most common mechanisms in PDT.

### 2.3. Types of PDT-Induced Cell Death

During the development of PDT, ROS such as H_2_O_2_, OH^−^, and O^2−^ are formed and can react with biological molecules such as lipids, DNA, and proteins to kill tumor cells through various cell death mechanisms (Figure 1). Research has indicated that there is no single pathway leading to cell death after PDT treatment. There are reportedly three main types of cell death: apoptosis, necrosis, and autophagy. They can occur either individually or simultaneously [23]. However, other types of cell death are also associated to PDT, such as necroptosis, pyroptosis, ferroptosis, cuproptosis, paraptosis, and parthanatos, among others.

#### 2.3.1. Apoptosis

Apoptosis is a type of programmed cell death that generally occurs in tumor cells in response to PDT treatment through several signaling pathways involving caspases, members of the Bcl-2 family of proteins, and apoptosis-inducing factors. However, in cases of PDT-induced cancer cell death events where the apoptotic pathway is not available (i.e., conditions where low levels of ATP are present in the cell), PDT may cause cancer cell death by inducing either the process of autophagy or necrosis [13,21].

#### 2.3.2. Autophagy

Autophagy, a form of programmed cell death, has the potential to either contribute to resistance to cancer treatments by suppressing cell death and allowing the cell to recover, or to promote susceptibility by facilitating the process of death [24]. It has been demonstrated in a number of PDT studies that the induction of cancer cell death by autophagy depends on the type of PS used, as well as the stage at which the cancer was diagnosed. Consequently, in PDT applications, it is essential to stimulate autophagy in parallel with apoptotic cell death in cancer cells to ensure their effective destruction, without the possibility of recovery.

PDT-induced necrotic cell death is a form of cell degeneration that is not scheduled. It affects large parts of cell populations. The key characteristics of this condition are cytoplasmic swelling, organelle destruction, and plasma membrane disruption. These factors ultimately lead to the release of intracellular contents and subsequent disintegration of cancer cells [13].

#### 2.3.3. Necroptosis

Necroptosis is a programmed form of cell death that is morphologically similar to necrosis. During this process, cells and their organelles undergo chromatin condensation and cell membrane rupture, releasing cellular contents into the extracellular space [25,26]. It has been demonstrated that the process of necroptosis, which is induced by PDT in the treatment of GB, is mediated by receptor-interacting protein 3 (RIP3). The initiation of this process is triggered by the production of singlet oxygen, resulting in RIP3-dependent necrosis. This process is characterized by the establishment of a pro-necrotic complex with RIP1 that differentiates itself from the components that comprise the conventional necrosome, such as that constituted by caspase—8 [27,28].

#### 2.3.4. Pyroptosis

Pyroptosis, a caspase-independent form of cell death, is characterized by cellular swelling and the appearance of numerous protrusions on the cell membrane prior to rupture [29]. It also involves the release of proinflammatory molecules, such as IL-18, IL-1B, and ATP. The activation of the inflammatory response is a hallmark of this process, enhancing its efficacy in combating tumors. The activation of this process is initiated by ROS. Recent studies have emphasized the underlying mechanisms of pyroptosis in PDT applications, thereby underscoring its potential as a treatment for cancer [30,31]. This process has received considerable attention in recent years due to its potential to cause immunogenic cell death (ICD) by activating antitumor immunity and counteracting immunosuppressive epithelial–mesenchymal transition (EMT) [32,33].

#### 2.3.5. Ferroptosis

Ferroptosis is iron-dependent and induced by the generation of ROS and lipid peroxides during PDT [34]. During this process, cellular labile iron storage increases and the Fenton reaction between H_2_O_2_ and ferric ions produces O_2_^−^ sustainably, increasing lipid peroxidation [35]. Ferroptosis is characterized by specific morphological, biochemical, and genetic features, and is regulated by multiple genes, primarily those associated with iron homeostasis and those involved in lipid peroxidation metabolism [24,35]. Observed morphological changes in cells include decreased mitochondrial and cristae size, increased membrane potential, and membrane rupture without chromatin condensation [36]. Biochemically, ferroptosis is initiated by glutathione depletion or the inhibition of glutathione peroxidase 4 (GPX4), a lipid repair enzyme. Intracellular glutathione plays a protective role against oxidative stress by catalyzing the reduction of phospholipid hydroperoxides. When reduced, this leads to an imbalance between oxidative damage and antioxidant defense, causing cell death [35,37].

#### 2.3.6. Cuproptosis

Cuproptosis is a recently identified copper-dependent cell death mechanism associated with the accumulation of copper in cells [38,39]. The phenomenon of cuproptosis in PDT for GB suggests that the presence of copper ions induces a process of mitochondrial decay and the subsequent accumulation of ROS. This, in turn, activates the AMP-activated kinase (AMPK) pathway, leading to the degradation of the programmed-death protein ligand 1 (PD-L1). The result of this is an enhancement of antitumor immunity and an improvement in therapeutic efficacy [38]. In this way, copper ions enhance the accumulation of therapeutic agents at tumor sites, facilitating the induction of cuproptosis through mitochondrial disruption and the generation of ROS during PDT. Another important aspect to consider is the development of a prognostic model related to cuproptosis that allows for the stratification of GB patients according to their response to cuproptosis. This, in turn, facilitates the development of personalized therapeutic strategies [39].

#### 2.3.7. Paraptosis

Paraptosis is a recently described form of programmed cell death that is characterized by swelling of the endoplasmic reticulum and mitochondria and vacuolization of the cytoplasm. In contrast to the process of apoptosis, paraptosis does not require the activation of caspases or DNA fragmentation [29,40,41]. The formation of vacuoles is attributable to the swelling of the endoplasmic reticulum and/or mitochondria, and these are surrounded by a single membrane [42]. Paraptosis is also linked to changes in calcium (Ca^2+^) and redox homeostasis, and it depends on members of the mitogen-activated protein kinase family [41,43]. The aforementioned characteristics render paraptosis an intriguing target for cancer therapy, particularly in the context of combatting apoptosis-resistant cells [44]. Recent studies have demonstrated that PDT can induce paraptosis in GB cells. This was evidenced by the extensive formation of cytoplasmic vacuoles, which are indicative of cellular stress responses [40,45].

#### 2.3.8. Parthanatos

Parthanatos is another form of programmed cell death. This process is distinguished by the excessive activation of the DNA damage response pathway, with a particular emphasis on poly(ADP-ribose) polymerase 1 (PARP1), and its independence from the conventional cell death pathway involving caspases [46]. PARP1 activation can be initiated by various DNA-damaging stimuli, including ultraviolet radiation, alkylating agents, and reactive ROS. PARP1 overactivation results in the accumulation of PAR polymers and cellular NAD+, as well as ATP depletion and energy collapse. This process of cell death is accompanied by mitochondrial depolarization, which contributes to the release of the truncated form of apoptosis-inducing factor (AIF) from the mitochondria [47]. Research findings suggest that it performs a significant function in the treatment of tumor cells including GB, especially when used in combination with PDT [48]. Genomic research has recently identified specific genes that can be used to make predictions about glioma patients’ prognoses and drug sensitivities. These genes include molecules such as CD58 and COL8A1 [49].

All mechanisms of PDT-induced cell death described above are highly dependent on several factors, including cell genotype, type of PS used, subcellular localization of PS, light dose used, and oxygen availability.

PDT can be applied locally to a specific region of the tumor by selectively illuminating the lesion without affecting normal tissue. PDT is therefore a more favorable option than radiotherapy and chemotherapy, which are known to be toxic. Over the past 40 years, PDT has been successfully used to treat skin cancer, Barrett’s esophagus, head and neck malignancies, lung cancer, and bladder cancer [13,50].

## 3. Photosensitizers (PSs) for PDT

The principal challenges inherent in the clinical approach to GB are undoubtedly associated with its diffusely infiltrative capacity. This capacity hinders the development of therapeutic agents capable of penetrating the BBB, and comes with the risk associated with tumor resection [51]. PDT is a light-based treatment modality that involves the administration of a PS to elicit a target response following photoactivation. It is widely acknowledged that a consequence of this treatment modality is the generation of ROS [52,53], which in turn induce tumoral cell death. This has resulted in the formulation of numerous strategies with the objective of enhancing the efficacy of PDT in the context of GB [54,55].

PSs are a key and important component of PDT. An ideal PS should be a chemically pure compound and easy to synthesize, have a homogeneous composition, efficiently generate ROS, selectively accumulate in target tissue, be harmless in the absence of radiation, absorb light in the long wavelength of the spectrum (600–850 nm, a range called the “phototherapeutic window”), be stable in solution, serum, or plasma, be easily eliminated from the body, and be economical to produce [56,57]. There are currently more than 1000 natural and synthetic PSs, which have been classified into generations [57,58,59] (Figure 2).

### 3.1. First-Generation Photosensitizers in PDT for GB

The first generation of PSs includes hematoporphyrin, photofrin (II) porphyrins, and their derivatives (Fotosan, Fotocan, and Fotofrin), (Figure 2). Despite their demonstrable efficacy, there are inherent limitations with regard to selectivity, and significant side effects on healthy tissue [7].

Their application to brain tumors was first demonstrated by Diamond et al. in 1972, who showed that the administration of hematoporphyrin followed by phototherapy had lethal effects on glioma cells in culture [60]. Furthermore, it resulted in massive destruction of subcutaneously transplanted porphyrin-containing gliomas in rats [60]. Although tumor growth was suppressed for a period of 10 to 20 days, subsequent reappearance of tumor cells was observed.

Subsequently, Perria et al. used an i.v.-injected hematoporphyrin derivative (HpD) as a sensitizing drug for the treatment of gliomas. The photodynamic process was initiated using a HeNe laser (632.8 nm). Following the surgical excision of the glioma, the residual tumor bed was exposed to the action of the laser, thus inducing necrosis of the remaining neoplastic cells [61].

Following the initial report by Perria, from 1980 to 1990, a number of groups from Italy [61,62], Australia [63,64], and the United States [65] documented the effective application of PDT with HpD in the management of glioma cases in modest patient populations.

In 1988, a phase I/II trial was conducted, involving 20 patients who were treated 25 times with PDT using a hematoporphyrin derivative and light at 630 nm (40–120 J/cm^2^). In 16 instances, PDT was administered subsequent to a single radiation dose of 4 Gy of fast electrons. The side effects (skin phototoxicity) observed in five patients were not only inconsequential but also did not have a detrimental effect on the patients’ quality of life. In certain cases, the survival rate was found to be considerably increased, thus providing evidence to suggest that PDT could be a valuable asset in the management of this particular group of patients [66]. In addition, other studies have shown an increase in survival in PDT-treated patients, mediated by hematoporphyrin derivatives [67].

In 2006, Muller conducted a randomized controlled trial of Photofrin^®^ PDT in patients with malignant glioma. The authors treated 112 patients with malignant gliomas, metastatic brain tumors, and meningiomas. The results showed that the treatment was safe as there were no adverse effects [68].

The initial research on photodynamic therapy has demonstrated encouraging outcomes in the management of gliomas. However, its effectiveness is constrained by the occurrence of non-specific accumulation of photosynthesizers in normal brain tissue. This may increase the risk of adverse effects, potentially disguising its safety and effectiveness [51,69,70]. The PSs had low chemical purity and could only be effectively activated at wavelengths below 640 nm, which limited their penetration into tissues. In addition, their long half-life resulted in skin hypersensitivity lasting several weeks, forcing patients to remain in a dark room for up to six weeks [43,71].

### 3.2. Second-Generation Photosensitizers in PDT for GB

Second-generation PSs undoubtedly have better advantages than first-generation PSs, including 5-aminolevulinic acid (5-ALA) and its esters, sodium taloporfin (LS11), benzoporphyrin derivative (BPD), temoporfin (mTHPC, Foscan), tin etiopururin (SnET2), and lutetium texaphyrin. Their higher purity, more efficient ROS production, and greater tumor selectivity mean that adverse effects are limited [51,70,72]. The PSs developed during the second generation were purely synthetic compounds consisting of macrocycles, such as chlorines, bacteriochlorines, and benzophorphyrins, and some common dyes, such as Rose Bengal, Eosin Y, and Methylene Blue (Figure 2) [57,59].

This second generation of PSs has been shown to possess phototoxic properties at longer wavelengths (600–800 nm). Furthermore, due to their excitation at lower energies (up to 20 J/cm^2^), these particles are able to penetrate tumoral tissues more profoundly. In addition, their elimination from the body was faster, resulting in fewer side effects and patients spending less time (less than two weeks) in a dark room. In recent decades, a number of PSs have been the subject of rigorous evaluation in a variety of clinical trials [51,73,74].

### 3.3. 5-ALA

5-ALA is an amino acid analogue and is the most commonly used second-generation PS for the treatment of malignant gliomas, including GB. Despite the fact that it is not an intrinsically fluorescent molecule, it is the precursor of the photosensitizing compound protoporphyrin IX (PpIX). It has been established that the process of PpIX generation in cells occurs through the heme biosynthetic pathway [54,75]. It has been observed that exogenous administration of 5-ALA leads to its accumulation in brain tumors, as well as in the surrounding infiltrating cancer cells. In the mitochondria, 5-ALA is converted into PpIX, and subsequently into heme, through the action of the enzyme ferrochelatase. It has been hypothesized that the low level of expression of this enzyme in neoplastic tissue may contribute to the accumulation of PpIX in gliomas and other tumor tissues [76]

It has been established that 5-ALA itself is a valuable tool for two main areas within the management of malignant gliomas. First, it is used for intraoperative real-time visualization (fluorescence-guided surgery) during surgical excision. Second, it contributes to improving treatment outcomes [74,77,78,79,80,81,82], as its employment in fluorescence-guided surgery has been shown to be associated with a higher rate of complete tumor resection and significantly longer progression-free survival [82]. In 2006, Stummer led a randomized, multicenter, phase III study in which 5-aminolevulinic acid fluorescence-guided surgery (5-ALA FGS) was used as an adjunct to surgery. This resulted in more complete tumor resections and improved overall progression-free survival at six months in patients [82].

At present, 5-ALA (also known as Gliolan©) has received FDA approval and is extensively utilized in the imaging of neoplastic tissue, including GB [83]. The following table provides a description of the clinical trials that have been approved for the management of GB, which are characterized by a dual orientation towards both visualization and therapeutic interventions.

### 3.4. Talaporfin (LS11, MACE, NPe6)

Thalaporfin is a chlorine derivative with a light absorption range in the 664 nm. The use of this agent has been demonstrated in patients with parenchymal changes in malignant brain tumors, with evidence of both effective and safe results in the management of these patients [84]. Further research, conducted using animal models, has demonstrated that PDT, utilizing sodium talaporfin (TPS), has the capacity to induce elevated levels of cell necrosis and diminished levels of tumor cell migration [26]. This observation was made in experiments in which C6 cells were injected into the frontal lobe of rats [85]. Concomitant observations have been reported by other researchers, who have highlighted that PDT in combination with talaporfin signifies a significant advancement in the therapeutic management of glioblastoma [86,87]. In 2019, a clinical study was conducted on 47 patients diagnosed with malignant gliomas. All patients received an intravenous injection of TPS at a dose of 40 mg/m^2^ 24 h prior to undergoing resection. During the surgical procedure, patients were exposed to a diode laser light emitting at a wavelength of 664 nm. This study indicates that TPS has the potential to serve as a valuable PS in two distinct applications: intraoperative fluorescence diagnosis and photodynamic therapy [88].

### 3.5. Phthalocyanines

Phthalocyanines constitute another group of second-generation PSs employed in PDT. The absorption spectra of these materials extend from 650 to 850 nm. It has been demonstrated that the utilization of phthalocyanines as PSs in PDT has the capacity to induce cell death, as evidenced by several cell models (T98G, MO59, LN229, C6, and U87-MG) [89,90,91,92].

The main disadvantage of second-generation PSs is their low water solubility. This property causes second-generation PSs to aggregate under physiological conditions, reducing the yield of ROS production. Their hydrophobic nature also limits their suitability for intravenous administration [93,94]. To address these limitations, liposomal formulations and modified NPs have been developed to improve water solubility and enhance their applicability.

### 3.6. Nanotechnology-Based Third-Generation Photosensitizers in PDT for GB

Third-generation PSs are currently under development, with a focus on synthesizing structures with a higher affinity and specificity, resulting in improved cellular internalization for target cells [6,7,23,58,59]. The range of delivery mechanisms encompasses a broad spectrum of entities, comprising polymers, lipids (e.g., liposomes), organometallic complexes, and nanospheres and nanocapsules [95,96,97]

As a scientific field of recent creation, nanotechnology is emerging as an alternative to improve the delivery, performance, and safety of current PSs and to overcome some of the challenges of PDT applied to cancer [98,99]. In recent decades, remarkable advances in medicine have been driven by nanotechnology due to the properties of nanomaterials. Although a precise definition of a nanomaterial has yet to be established, the International Union of Pure and Applied Chemistry (IUPAC) describes it as a substance whose particles’ size ranges from 1 to 100 nm [100,101,102]. Materials designed at the nanometer level exhibit novel properties that are not observed in bulk materials of the same composition. For instance, increasing the surface-to-mass ratio of nanomaterials can significantly impact their reactivity. These new properties can be exploited to overcome limitations associated with traditional therapeutic and diagnostic agents [100,101]. Significant progress has been made in developing various nanomaterials for use primarily in drug delivery, disease diagnosis, and therapy. These materials can improve drug stability and selectivity, reduce drug-induced side effects caused by drugs, provide controlled and targeted drug release, and enhance therapeutic efficacy [103,104,105]. In general, most of these functional nanomaterials can be used directly as PSs or as carriers of PSs, significantly improving the efficacy and safety of PDT [106,107,108]. Using nanomaterials as drug carriers offers several advantages, such as improved solubility and stability, controlled release, and tumor-specific delivery of PSs. Nanoparticle-based systems have also been used to deliver both PSs and anticancer agents together, in order to achieve combined anticancer therapy [106].

Combining photosensitizers with nanomaterials can improve the efficacy of photodynamic therapy and eliminate its side effects. Using nanoparticles enables a targeted approach focusing on specific receptors and increasing the selectivity of photodynamic therapy [109,110,111,112,113,114,115,116]. Additionally, they may prevent the early release and inactiavtion of PS, and may also improve the solubility of hydrophobic PS molecules.

Nanomaterials used in photodynamic therapy consist of numerous structures constructed from organic and inorganic materials (Figure 3). Among organic nanomaterials are liposomes, dendrimers, lipid-based nanoparticles, micelles, and polymeric nanoparticles. As for inorganic nanomaterials, these include gold and silver nanoparticles, silica nanoparticles, quantum dots, carbon materials (nanotubes, fullerenes, graphene, carbon dots), magnetic nanoparticles, metal nanoparticles, metal oxide nanoparticles, and semiconductor nanoparticles. These materials have shown significant results in terms of stability, biodistribution, and cytotoxic and therapeutic efficacy [117,118,119,120,121,122,123].

A significant benefit of utilizing NPs is their capacity to safeguard PSs from premature degradation, thereby extending their stability and facilitating their accumulation within tumor tissue [72]. In addition, PS-loaded NPs can be efficiently transported to the tumor site. In the case of malignant gliomas, these PSs facilitate passage through the BBB to target cells [124,125,126].

The use of NPs loaded with a photosensitizer (AlClPc) was used for photodynamic therapy of GB. The results demonstrated elevated levels of ROS production, accompanied by a decline in the viability and proliferation of tumor cells within a murine model. The authors observed a significant decrease in the total tumor area and no evidence of systemic toxicity [127]. In 2023, Comincini et al. demonstrated that the PS berberine, when incorporated into NPs, could induce apoptosis in cancer cells. The study also revealed that this process did not elicit a cytotoxic response in healthy tissue [128].

## 4. TiO_2_- and ZnO_2_-Based Nano-Photosensitizers

In the recent literature, the employment of TiO_2_ and ZnO as PSs has emerged as a subject of considerable interest in the context of PDT [117,129]. TiO_2_NPs and ZnONPs have been shown to act as PSs in PDT, due to their capacity to generate ROS under light irradiation. The generation of ROS in these cases has been demonstrated to be a highly effective mechanism in the destruction of cancer cells. It is evident that both types of NPs possess the capacity to function as PSs in isolation or to undergo modification with other chemicals, thereby enhancing their ROS production and therapeutic effects [130,131].

### 4.1. TiO_2_

TiO_2_, otherwise known as titania, is the most prevalent semiconductor amongst metal oxides. It is extensively used as a photocatalyst due to its capacity to generate substantial quantities of ROS when exposed to ultraviolet light [132,133]. The photocatalytic activity exhibited by the material is attributable to charge separation processes that occur both in its bulk and on its surface. The most well-known polymorphic structures of TiO_2_ are anatase, brookite, and rutile (Figure 4); all of these structures are active under UV irradiation, as they have a forbidden band (Eg) of approximately about 3.3 eV [134]. TiO_2_ can be synthesized in a variety of ways, including sol–gel, hydrothermal, green chemistry, and microwave methods [135].

Titania possesses relevant properties, including low cost, availability, and chemical stability, high photostability and photocatalytic activity, and excellent biocompatibility [136,137]. Activation of TiO_2_ by UV light has been demonstrated to generate electrons and holes in the conduction and valence bands, respectively (Figure 4). This process ultimately results in the generation of ROS via the redox reactions of oxygen/water molecules on the TiO_2_ surface [136,137]. It is for this reason that TiO_2_ has attracted attention for its application as a PS in PDT because it produces reactive oxygen species such as hydroxyl radicals, hydrogen peroxides, and superoxides, in aqueous media and under irradiation with UV light [138]. The toxic effect of this on cancer cells is remarkable, causing severe oxidative stress and consequently apoptosis [136,139]. The application of TiO_2_-NPs as a PS in PDT against various types of cancer cells (bladder, cervical, non-small-cell lung, human skin, breast, leukemic HL60, and GB cancers) has been documented [140,141,142,143,144,145]. Although TiO_2_ is a potential ROS-generating agent, it can only be photo-activated with UV light (UV light is not suitable for PDT due to its limited tissue penetration ability and inherent deleterious effects), which limits its use in PDT as the approved therapeutic window is at wavelengths of 700–1100 nm. In light of the aforementioned facts, TiO_2_ has been modified to absorb long-wavelength light in the visible or near-infrared (NIR) range. TiO_2_ has been doped with a variety of metal and non-metal ions and combined with a range of dyes [146,147,148,149]. Among organic dyes, porphyrins and phthalocyanines are the most commonly used in combination with TiO_2_. This combination is used to obtain a PS with better absorption in the visible region [150]. To enhance the uptake of TiO_2_ NPs by cancer cells, specific ligands, such as monoclonal antibodies, peptides, and aptamers that bind to specific proteins or surface antigens, can be coupled to their surface. For instance, the folate receptor (FR) has been reported to be overexpressed on the surface of many tumor cells, whereas it is underexpressed on the surface of normal cells [150,151].

#### Toxicity of TiO_2_NPs

The primary objective of employing TiO_2_NPs as a drug carrier system is to enhance the effectiveness of the treatment while concomitantly minimizing the incidence of adverse reactions [152]. However, the potential hazardous effects of TiO_2_NP are not yet fully understood [153].

Several studies have suggested that TiO_2_NPs can enter the human body through various exposure routes, such as inhalation, injection, skin contact, and intestinal absorption, and travel to different regions of the body through the blood or lymph and subsequently generate some toxicity [154]. It has been demonstrated that vital organs, including but not limited to the heart, stomach, spleen, kidneys, liver, gastrointestinal tract, reproductive system, lungs and brain, are sites of nanoparticle accumulation [155,156]. In this sense, before using this kind of NP in the treatment of GB or other brain tumors, its harmful potential for other vital organs, such as the heart or gut tract, must be considered.

As demonstrated by several researchers, the concentration and size of TiO_2_NPs have been shown to play an important role in determining their toxicity. Additional factors must be taken into consideration, including the duration of exposure and the crystalline structure. As nanosized chemical compounds have been demonstrated to exhibit greater potency than their bulk counterparts, concerns have been raised regarding exposure to TiO_2_NPs [157,158,159].

Regarding the CNS, research has demonstrated that the capacity of TiO_2_NPs to cross the BBB is a desirable effect for therapeutic purposes; however, these NPs can impair its permeability beyond the intended level. This phenomenon can be attributed to the degradation of tight junction proteins, resulting in an accumulation of TiO_2_NPs within the brain [153,160]. Consequently, the induction of oxidative stress, apoptosis, neuroinflammation, and neuronal degeneration can be triggered [161]. Furthermore, the increase in BBB permeability, due to the accumulation of TiO_2_NPs, induces the signaling of proinflammatory pathways mediated by IL-1R and IL-6 [155,162,163]. This may constitute a concern when using TiO_2_NPs as PS carriers, since the inflammatory response may affect healthy surrounding tissue.

Furthermore, the neurotoxic damage of TiO_2_NPs is also reflected in impaired neurogenesis, reduced neurite growth, and impaired neurotransmitter metabolism, especially dopamine and glutamate. In addition, the cytotoxic effect on glial cells has been observed [164,165].

### 4.2. ZnO

ZnO is an n-type semiconductor with a wide bandgap energy (Eg) of 3.37 eV. This is due to its chemical structure, which contains a large number of intrinsic defects, such as zinc interstitials and oxygen vacancies [166]. It is also considered biocompatible, being classified as a “GRAS” (generally recognized as safe) substance by the Food and Drug Administration (FDA). Other properties of ZnO include high photostability and chemical stability. Additionally, it possesses paramagnetic properties, resulting in a wide range of radiation absorption and a high electrochemical coupling coefficient. ZnO can assume different chemical structures known as wurtzite, rock salt, and cubic blende (Figure 5) [167]. ZnO can be synthesized into a variety of nanostructures, including nanotubes, nanowires, nanowires, nanowafers, nanoflowers, quantum dots, and nanobelts (Figure 5) [168,169,170,171,172,173,174]. In the field of biomedicine, it is used as a vehicle for transporting drugs and anticancer agents. ZnO absorbs ultraviolet (UV) radiation and generates reactive oxygen species (ROS) such as OH^−^, O_2_^−^, ^1^O_2_, and H_2_O_2_. These ROS can react with adsorbed surface water, hydroxyl, and oxygen molecules [53] (Figure 5). The intracellular production of ROS has the potential to exert a cytotoxic effect, making them a valuable tool in the fight against cancer. Indeed, excessive intracellular ROS production can lead to a number of detrimental outcomes, including oxidative stress, alterations to the cell cycle, and the promotion of cell death by either apoptosis or autophagy. Consequently, numerous studies have proposed the use of photoexcitation of ZnO nanostructures to generate intracellular ROS as an effective therapeutic strategy for PDT.

ZnO-based nanomaterials have also been investigated as PSs in PDT due to their high surface-to-volume ratio [175,176,177,178]. ZnO nanoparticles have been shown to have a photosensitizing effect on hepatocellular carcinoma cells (HepG2) [179]. In addition, ZnO nanostructures are used for drug delivery and diagnostic purposes.

Like TiO_2_, the rapid recombination of photoinduced carriers under UV light and the low absorption of visible light limit the application of ZnO as a PS. Charge carrier recombination can be suppressed by trapping photoinduced electrons or holes by inducing surface vacancy defects and constructing Schottky junctions [175]. Therefore, the low optical absorption of ZnO and TiO_2_ in visible light can be enhanced by means of oxygen vacancy formation, noble metal doping, nonmetallic doping, and modification with carbon nanomaterials, amongst other methods [175].

#### Toxicity of ZnONPs

Similar to TiO_2_NPs, ZnONPs are extensively employed as raw materials in the pharmaceutical and cosmetic industries, as pigments in the paint, concrete, and rubber industries, and as UV filters in textiles, agricultural products, medical products, and biological products [180]. Humans are frequently exposed to ZnO nanoparticles because they can enter organisms through multiple routes, such as the respiratory tract, digestive system, and injectable routes [181]. Following internalization, these particles are capable of accessing various tissues and organs, including but not limited to the liver, kidneys, and brain. This can result in an alteration of the immune system, leading to an inflammatory response [182,183,184,185,186,187].

In a study conducted on human melanocytes, the cytotoxic potential of ZnONPs and TiO_2_NPs was evaluated with and without exposure to UV radiation. The results demonstrated that ZnONPs are more cytotoxic, especially when exposed to UV light. The observed cytotoxic effect is also related to the concentration and size of the NPs. It was also demonstrated that various intracellular processes, such as cell membrane integrity, proliferative processes, and the induction of morphological changes, in cells at the ultrastructural level, particularly in mitochondria, are affected by ZnONPs (5–12.5 ppm) [188]. Similar results have been observed in human gingival fibroblast cells and intestinal cells [189,190].

The cytotoxic effects of ZnONP accumulation are highly relevant. The literature indicates that exposure to this type of NP results in the distribution of Zn in the kidneys, liver, and lungs. However, over time, the concentration of Zn in these organs decreases, while it increases in the brain [191]. Zn accumulation in the brain induced oxidative stress and neuronal damage, and affected spatial learning and memory [192]. The neurotoxic effects of ZnONPs have been demonstrated to be mediated by a number of factors, including apoptosis in neuronal, necrosis, and ferroptosis of both neuronal and glial cells [118,193,194,195].

Other mechanisms include brain oxidative stress due to a deregulation in the activity of the superoxide dismutase (SOD), catalase (CAT), glutathione-S-transferase (GST), and acetylcholinesterase (AChE) enzymes, promoting neuroinflammation [196]. Impaired autophagy, cytoskeletal damage, mitochondrial dysfunction, altered neurotransmitter metabolism, and synaptic transmission are other toxic mechanisms caused by the accumulation of ZnOPs [183,197,198,199]. Therefore, the challenge is to design nanoparticles that ensure these effects are selectively confined to the tumor and its immediate peritumoral microenvironment.

### 4.3. Protein Corona in TiO_2_NPs and ZnONPs

Nanomaterials (NMs), such as TiO_2_NPs and ZnONPs, intended for medical use will come into contact with biological fluids. Therefore, these NPs may interact with biological components such as lipids, proteins, nucleic acid fragments, cell debris, and cells. The most studied molecules that form these complexes are proteins known as a “protein corona” (PC), which can form a layer on the surface of NMs [200]. It has been established that this coating, which forms around nanoparticles, comprises two layers: a hard corona (HC) of proteins that strongly bind to the nanoparticle surface and a soft corona (SC) of proteins that bind less strongly [201]. The formation of this PC affects the bio-identity of the NM, resulting in modifications to its behavior, fate, and pharmacological profile. Furthermore, it has been demonstrated that parameters such as particle circulation time, distribution, and cellular toxicity are contingent on the presence and composition of the NPs protein corona.

A study was conducted to characterize the formation of PCs on TiO_2_NPs based on their main properties, and to investigate the possible relationship between the formation of these coronas and the cytotoxicity induced by the nanoparticles in human lung cells in vitro. The results showed that the size and surface charge of the nanoparticles did not determine protein corona formation, and there was no clear impact of the shape or agglomeration state of the NPs. Additionally, no evident relationship was found between the protein corona of the nanoparticles and the adverse effects they caused [202].

Poulsen et al. used proteomic results to ascertain that a unique protein corona is formed on the surface of TiO_2_NPs in lung fluid. Furthermore, the expression levels of three proinflammatory cytokines (interleukin 6 (IL-6), tumor necrosis factor alpha (TNF-α), and macrophage inflammatory protein 2 (MIP-2)) were measured. The study revealed that the corona formed from lung fluid exhibited elevated levels of these cytokines in comparison to both plain TiO_2_ nanoparticles and coronas formed from serum or albumin [203].

The formation of PCs on ZnONPs when in contact with biological fluids has also interestingly been studied. Bhunia, A. K. et al. studied the interaction between ZnONPs and human hemoglobin (Hb). The Hb formed a “hard corona” on the surface of the ZnONPs in a very short time, and the unfolding of Hb occurred over a long time. The ZnO NPs were completely covered by Hb with a shell thickness of 6 nm. Tryptophans and heme–porphyrin moieties of Hb were the major binding sites for ZnONPs. Electrostatic interaction, along with the hydrophobic interaction between ZnO NPs and Hb, is responsible for the conformational change in Hb due to the increase in the percentage of beta-sheets, together with a decrease in alpha-helices [204].

It has been demonstrated in further studies that protein-coated ZnONPs in serum-containing media exhibit reduced levels of toxicity and an augmented release of extracellular ions in comparison with the same particles that have been incubated in serum-free media [205].

## 5. Cancer Therapy Using Nanomaterials

Since the 1980s, cancer treatment has evolved significantly. We have moved away from broad-spectrum cytotoxic chemotherapeutics towards the development of targeted therapies and the inhibition of specific cancer pathways [206]. It is important to note that radiotherapy and chemotherapy are not very effective at distinguishing between cancerous and normal cells, which can lead to serious side effects. This underscores the emergence of targeted drug delivery systems utilizing nanomaterials as a promising approach for cancer treatment.

Active delivery strategies have been shown to enhance the intracellular concentration of therapeutic agents in cancer cells, while sparing normal cells from cytotoxic effects [207,208,209]. The conjugation of different elements to the surface of nanomaterials (e.g., antibodies, peptides, and growth factors) containing chemotherapeutic drugs has been demonstrated as a means to achieve this [210,211,212,213,214].

### TiO_2_ and ZnO Nanoparticles in PDT for GB

The photochemical properties of TiO_2_ and ZnO nanoparticles in PDT for GB treatment are promising due to their ability to generate ROS upon light activation, which can induce cancer cell death. It has been demonstrated that both TiO_2_ and ZnO nanoparticles are highly effective as PS in PDT. In addition, these nanoparticles have demonstrated their capacity to target and destroy GB cells with greater precision.

The potential of TiO_2_NPs and ZnONPs as therapeutic agents in the treatment of GB is supported by a number of in vitro studies; it also has potential for drug delivery [215,216,217]. As shown in Figure 6, these NPs have been shown to effectively cross the BBB, thereby targeting GB cells while minimizing systemic toxicity (Figure 6).

In 2014, Zhang et al. conducted a comparative analysis of TiO_2_NPs and ZnONPs as PSs in PDT. The hepatocarcinoma cells (SMMC-7721) were incubated with both NPs, after which it was demonstrated that upon UV irradiation, ROS were generated. The ROS were determined by the dichloro-dihydro-fluorescein diacetate (DCFH-DA) method. Furthermore, the expression levels of apoptosis-related genes such as Bax, Bcl2, and Caspase 3 were determined, thereby demonstrating that both NPs are capable of inducing ROS production as well as caspase-dependent apoptosis, resulting in anticancer activity [131].

A key challenge in implementing TiO_2_NPs and ZnONPs as PSs in PDT is the necessity of a wavelength within the 315–400 nm range, which falls below the permitted therapeutic window. Consequently, methodologies have been formulated to enhance the near-infrared (IR) absorption wavelength. One such strategy involves the functionalization of NPs with molecules, such as folic acid (FA). FA is widely regarded as the optimal ligand for the targeted delivery of imaging and therapeutic agents to cancerous tissues and areas of inflammation [218,219]. The use of FA as a ligand has gained increased attention due to several key factors: firstly, its small size, availability, and simple conjugation chemistry, make it highly versatile, and secondly, its high affinity for folate receptors, which are overexpressed by many cancer cells but have limited distribution in normal tissues. These advantages make it a highly effective solution [220,221,222,223,224]. FA is a water-soluble vitamin that readily conjugates to supramolecular and macromolecular structures. It has been used as a vector to target cancer cells due to its high affinity (*K**d*∼10–10 mM) for their corresponding folate receptors (FRs: FR*α*, FR*β*, and FR*γ*) to mediate cellular uptake of FA [224,225,226,227,228,229,230,231]. When FA conjugates bind to RFs, they can be transported into the cell by RF-mediated endocytosis (Figure 6). For instance, the elevated surface area, augmented pore volume, and spherical morphology of hollow mesoporous nanoparticles (HMNPs) make them effective nanocarriers in the domains of drug delivery and PDT [232,233].

In this sense, Uribe-Robles et al. synthesized hollow TiO_2_ nanospheres (HTiO_2_NS), which were then fused with FA to enhance cell selectivity. This was followed by the subsequent fusion of HTiO_2_NS with zinc (II) tetranitrophthalocyanine (ZnPc), allowing for light absorption within the visible spectrum [150]. The cytotoxic potential of fusionalized HTiO_2_NS was assessed in GB cells (M059K), thereby demonstrating the capacity of these nanostructures to generate ROS. As indicated by the findings of other studies, TiO_2_-ZnPc nanoparticles which have been modified with FA have the capacity to target GB cells selectively, offering a more precise and potentially less toxic treatment option. The outcomes of this study further substantiate that TiO_2_-FAZnPc exhibits biocompatibility and is exclusively activated by light irradiation, thereby inducing the generation of ROS. This property lends weight to its potential as a treatment for this form of cancer by demonstrating its PS capacity [234].

## 6. Conclusions

In recent decades, there has been an accumulation of knowledge regarding the function and use of PDT, as well as its application in the treatment of malignant brain tumors, such as GB. The efficacy of PDT as a tumor treatment modality is predicated on its highly selective nature, which means a low incidence of adverse effects. Consequently, this treatment has been shown to enhance patients’ quality of life. Furthermore, the utilization of nanotechnology as a delivery system for these PSs has demonstrated substantial progress, as evidenced by a notable reduction in systemic toxicity. However, it is imperative to advance the design of more effective NPs that have the ability to penetrate the BBB in order to improve therapeutic outcomes.

The possibility to conjugate and functionalize TiO_2_ and ZnO nanoparticles is an emerging frontier in glioblastoma photodynamic therapy (PDT), combining selective tumor targeting with intrinsic photoreactivity. Beyond their demonstrated efficacy in cellular models, the next critical step is to develop nanosystems that respond to near-visible wavelengths. This would enable systemic intravenous administration or localized perioperative application after tumor resection to eliminate residual tumor cells in vivo. This could overcome the limited tissue penetration that occurs when PSs are activated with ultraviolet light. Advancing this field demands a concerted research effort to engineer nanostructures with optimized photosensitizer properties, precise biodistribution, and robust safety profiles to establish a foundation for clinical translation.

## Figures and Tables

**Figure 1 pharmaceutics-17-01132-f001:**
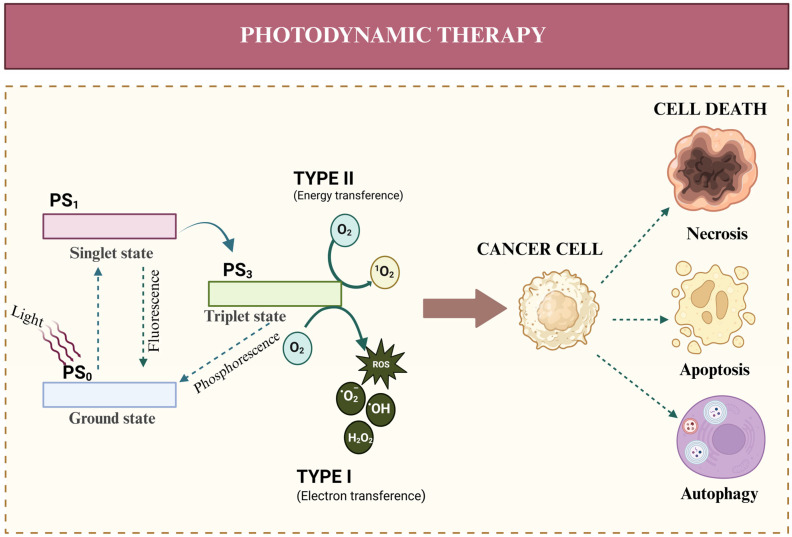
The process through which ROS are produced through two types of reactions (energy transference or electron transference) from the light irradiation of a PS. The three main kinds of cell death induced by PDT. PS_0_ is the photosensitizer in its basal state, PS_1_ is the photosensitizer in its excited singlet state, PS_3_ is the photosensitizer in its triplet excited state, ROS is reactive oxygen species, DNA is deoxyribonucleic acid, and ER is endoplasmic reticulum.

**Figure 2 pharmaceutics-17-01132-f002:**
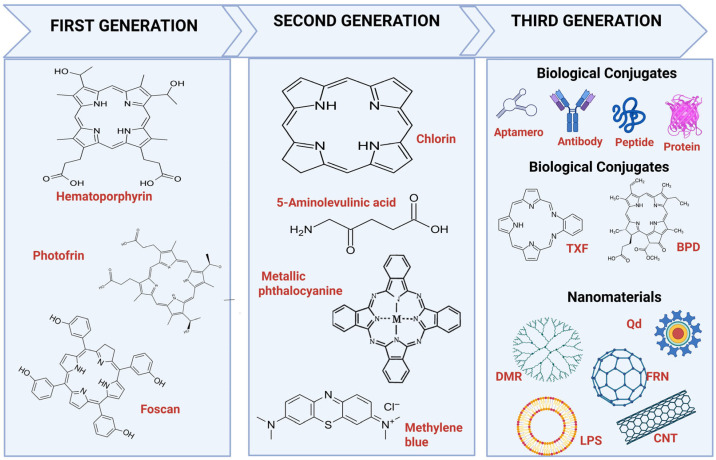
Examples of the different generations of photosensitizers that have developed over time. TXF, toxoflavin; BPD, benzoporphyrin derivative; DMR, dendrimer; Qd, quantum dot; FRN, fullerene; LPS, liposome; and CNT, carbon nanotube.

**Figure 3 pharmaceutics-17-01132-f003:**
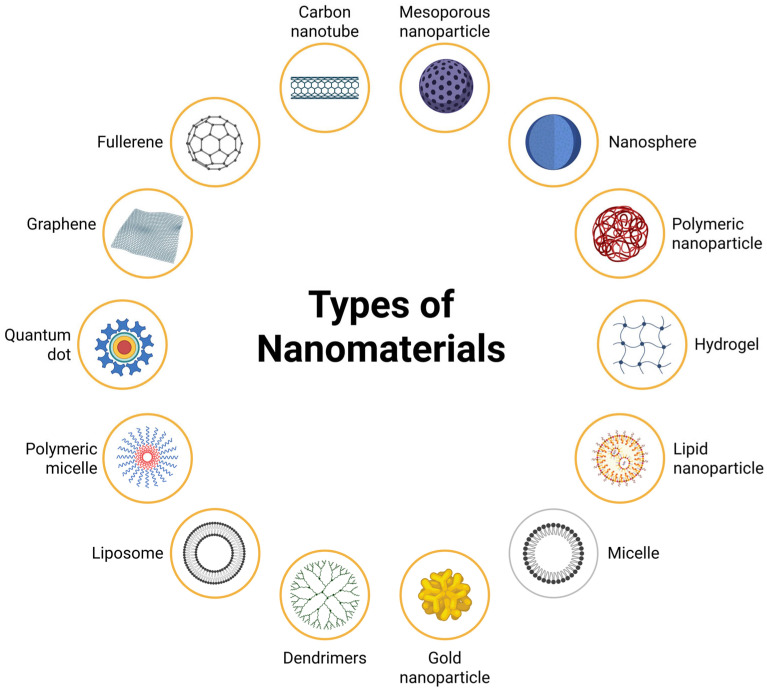
Main types of nanomaterials: organic nanomaterials include polymeric nanoparticles, lipid nanoparticles, polymeric micelles, dendrimers, micelles, and liposomes. Inorganic nanomaterials include mesoporous silica nanoparticles, metal nanospheres, gold nanoparticles, and quantum dots. Carbon-based materials include graphene, fullerene, and carbon nanotubes.

**Figure 4 pharmaceutics-17-01132-f004:**
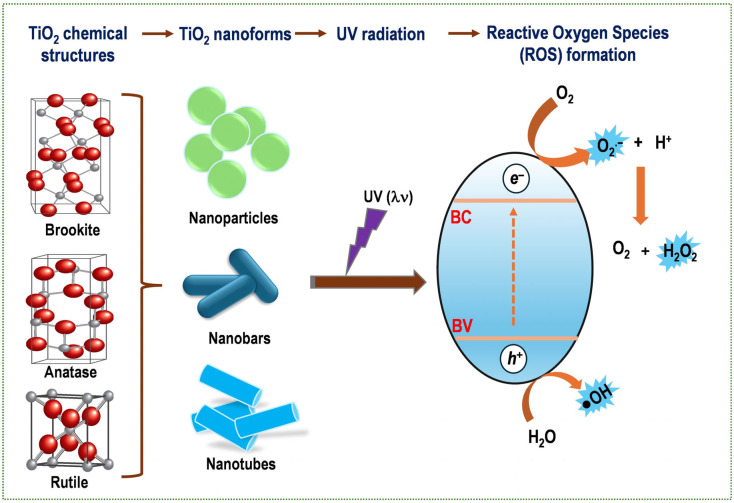
The three main crystalline structures of TiO_2_, nanoforms that TiO_2_ can take, and the formation of ROS by irradiation of TiO_2_ with UV light.

**Figure 5 pharmaceutics-17-01132-f005:**
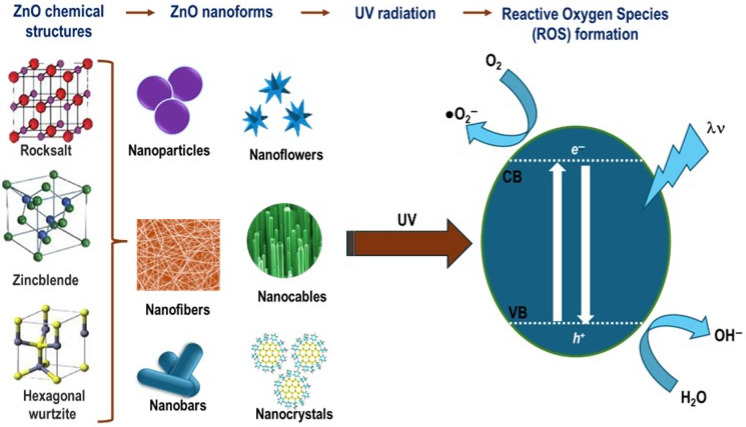
Main crystalline structures of ZnO, nanoforms of the zinc oxide and ROS formation by irradiation of ZnO with UV light. VB, valence band; CB, conductance band.

**Figure 6 pharmaceutics-17-01132-f006:**
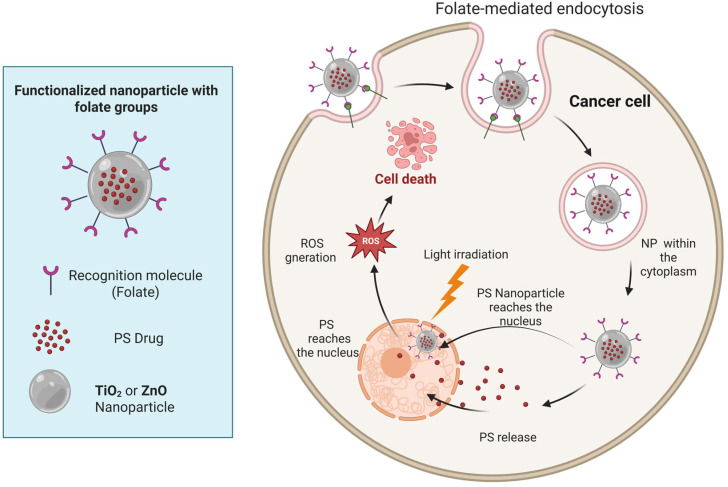
TiO2 and ZnO NPs have the potential to be functionalized with folate groups (and/or other conjugates) so that they can be taken up by a cancer cell through folate-receptor-mediated endo-cytosis. Once inside the cell, the NP can release the PS, or it can act as a PS, generating ROS through light irradiation, which damage the nucleus and other cellular organelles, inducing cell death.

## Data Availability

Not applicable.

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
