# Peer review of "Photodynamic Therapy for Glioblastoma: Potential Application of TiO2 and ZnO Nanoparticles as Photosensitizers"

_pharmaceutics, 2025, doi:10.3390/pharmaceutics17091132_

Round 1
Reviewer 1 Report
Comments and Suggestions for Authors
The study by Ortíz-Islas et al. critically evaluates the rationality of use the combination of photosensitisers with nanomaterials (TiO2 and ZnO2-based nano-photosensitisers) for boosting photodynamic therapy in the management of glioma. Overall, the manuscript is within the scope of Pharmaceutics journal.
However, in my view, the current version has certain shortcomings that could diminish the reader's interest. Sections 6 and 7 are written in general and do not present substantial novelty and the authors did not fully elucidate the role of TiO2 and ZnO nanoparticles as photosensitizers in glioma treatment.
Please find below my other suggestions that may improve the manuscript.
Major critiques:
- The authors should clearly specify whether they are discussing gliomas or glioblastomas, as these are not equivalent terms.
- Please provide the full name of the abbreviation (e.g., photodynamic therapy (PDT), reactive oxygen species (ROS)) upon first mention in the text, and then use only the abbreviation (PDT, ROS) thereafter.
Please unify the abbreviation for photosensitizer – PS or FS?
- Section 2 (Photodynamic Therapy (PDT)) requires substantial revision.
The current description of photodynamic reactions is overly superficial and fails to provide essential details. Moreover, Figure 1 lacks critical annotations and the figure legend should clarify them (ex. the terms "PSes" and "PSet" are introduced without definition).
The section also contains an incomplete coverage of PDT-induced cell death mechanisms: only three forms (apoptosis, necrosis, autophagy) are mentioned, while the authors themselves reference an article that significantly extends this list (e.g., ferroptosis, necroptosis, parthanatos, pyroptosis etc).
- Please provide more detailed descriptions for other figures. The current versions are extremely uninformative and force readers to search for information in the main text.
+ Figure 2 – what is FBD?
- Expand the information and fully explore the prospects of using TiOâ‚‚ and ZnO nanoparticles as photosensitizers for gliomas. Why do they have potential specifically for gliomas? Support your hypothesis with existing studies.
Minor comments:
There are minor grammatical errors and many misprints and stylistic inaccuracies; therefore, the manuscript needs to be proofread carefully.
Lines 203-222, 289-293 – apparently, extra text.
Author Response
Reviewer 1
The study by Ortíz-Islas et al. critically evaluates the rationality of use the combination of photosensitisers with nanomaterials (TiO2 and ZnO2-based nano-photosensitisers) for boosting photodynamic therapy in the management of glioma. Overall, the manuscript is within the scope of Pharmaceutics journal.
However, in my view, the current version has certain shortcomings that could diminish the reader's interest. Sections 6 and 7 are written in general and do not present substantial novelty and the authors did not fully elucidate the role of TiO2 and ZnO nanoparticles as photosensitizers in glioma treatment.
Please find below my other suggestions that may improve the manuscript.
Thank you very much for taking the time to review this manuscript. Below you will find detailed responses and corresponding corrections by red text throughout the manuscript.
Major critiques:
Comment 1. The authors should clearly specify whether they are discussing gliomas or glioblastomas, as these are not equivalent terms.
Response 1: We agree with this comment. We have therefore made the suggested changes.
Comment 2. Please provide the full name of the abbreviation (e.g., photodynamic therapy (PDT), reactive oxygen species (ROS)) upon first mention in the text, and then use only the abbreviation (PDT, ROS) thereafter.
Response 2: Thank you for pointing that out. We have made the requested changes.
Comment 3. Please unify the abbreviation for photosensitizer – PS or FS?
Response 3: I am grateful to you for highlighting this matter. The standardisation of the abbreviation for photosensitiser has been implemented.
Comment 4. Section 2 (Photodynamic Therapy (PDT)) requires substantial revision. The current description of photodynamic reactions is overly superficial and fails to provide essential details. Moreover, Figure 1 lacks critical annotations and the figure legend should clarify them (ex. the terms "PSes" and "PSet" are introduced without definition).
Response 4: Thanks for the suggestion. An additional section has been incorporated in order to provide a more detailed description of photodynamic reactions (Pages 3-4). Furthermore, as recommended, an additional correction has been made to legend figure 1. (page 4)
Comment 5. The section also contains an incomplete coverage of PDT-induced cell death mechanisms: only three forms (apoptosis, necrosis, autophagy) are mentioned, while the authors themselves reference an article that significantly extends this list (e.g., ferroptosis, necroptosis, parthanatos, pyroptosis etc).
Response 5: I am grateful to you for highlighting this matter. In order to provide a comprehensive description of the other types of cell death related to PDT, the requested information has been incorporated into the text (pages 5-7).
Comment 6. Please provide more detailed descriptions for other figures. The current versions are extremely uninformative and force readers to search for information in the main text.
Response 6: We agree with this comment. We have therefore made the suggested changes by correcting the leyend figures.
+ Figure 2 – what is FBD?
Thank you very much for the comment. Figure 2 has been updated to reflect these changes.
Comment 7. Expand the information and fully explore the prospects of using TiOâ‚‚ and ZnO nanoparticles as photosensitizers for gliomas. Why do they have potential specifically for gliomas? Support your hypothesis with existing studies.
Response 7: I would like to express my sincere gratitude for your insightful commentary. The present study has expanded the information regarding the use of TiO2NPs and ZnONPs as photosensitizers for GB.
Minor comments:
There are minor grammatical errors and many misprints and stylistic inaccuracies; therefore, the manuscript needs to be proofread carefully.
Lines 203-222, 289-293 – apparently, extra text.
Thank you for your comments. We've done a full review to fix the problems.
Lines 203-222, 289-293 have been eliminated

Reviewer 2 Report
Comments and Suggestions for Authors
In my opinion, the work entitled “Treatment of glioma by photodynamic therapy using TiO2 and ZnO nanoparticles as photosensitizers” presents interesting results and is suitable for the journal Pharmaceutics, considering the scope of both the work and the journal. Nevertheless, the manuscript should be significantly improved before publication.
First of all, the authors should remove the fragments of text taken from the MDPI form. Secondly, I am not convinced that the title of the review fully corresponds with the content of the work.
In my opinion, the focus of the work is not properly placed on the application of TiO2 and ZnO in the treatment of glioma. The authors mixed issues related to different types of photosensitizers. I think that too much attention was paid to the first two classes, whereas the aspects related to photodynamic therapy using both metal oxides are insufficiently covered. I suggest reorganizing the structure of the text and not including the fragments related to the first and second classes of photosensitizers.
The issues related to the biological activity of TiO2 and ZnO should be more thoroughly developed. In many literature reports, one can find information regarding the cytotoxicity of both modified metal nanoparticles on human cells. The authors should include this information in the review, considering the concentration of nanoparticles and their surface modification. Moreover, they should discuss the issue of biocompatibility of both materials, taking into account their surface properties.
The issues of protein-corona formation on the surfaces of both metal oxides should be discussed in detail, especially in comparison to the use of these materials for the delivery of other biologically active materials.
The term nanomaterial should be defined according to the IUPAC definition (page 5).
Fig. 3: The captions should be improved – Liposomes instead of Kiposomes.
I suggest using the abbreviation TiO2Ps to indicate TiO2 in the form of particles.
Reactive oxygen species are described with their forms on page 10, although they are discussed earlier. I suggest defining reactive oxygen species in the introduction.
What does “chemically pure” mean? (page 4)
The examples of application of TiO2 and ZnO in photodynamic therapy should be discussed in detail. I suggest adding relevant literature reports describing these issues.
I think that more details related to the synthesis and preparation of both types of metal oxides should be included. The authors should also discuss how the preparation procedures and purification methods of these particles influence their biological activity.
Summarizing, I suggest improving this review to increase its quality.
Author Response
Reviwer 2
In my opinion, the work entitled “Treatment of glioma by photodynamic therapy using TiO2 and ZnO nanoparticles as photosensitizers” presents interesting results and is suitable for the journal Pharmaceutics, considering the scope of both the work and the journal. Nevertheless, the manuscript should be significantly improved before publication.
The authors are immensely grateful for the time taken to review this manuscript. The following section contains detailed responses and the corresponding corrections, which are indicated by red text throughout the manuscript.
Comment 1. First of all, the authors should remove the fragments of text taken from the MDPI form.
Response 1: We agree with this comment. We have removed all fragments of text taken from MDPI template
Comment 2. Secondly, I am not convinced that the title of the review fully corresponds with the content of the work.
Response 2: Thank you for the comment, the title has been changed.
Comment 3. In my opinion, the focus of the work is not properly placed on the application of TiO2 and ZnO in the treatment of glioma. The authors mixed issues related to different types of photosensitizers. I think that too much attention was paid to the first two classes, whereas the aspects related to photodynamic therapy using both metal oxides are insufficiently covered. I suggest reorganizing the structure of the text and not including the fragments related to the first and second classes of photosensitizers.
Response 3: The article was restructured, now text on the application of TiO2 and ZnO in PDT for glioblastoma was added.
Comment 4: The issues related to the biological activity of TiO2 and ZnO should be more thoroughly developed. In many literature reports, one can find information regarding the cytotoxicity of both modified metal nanoparticles on human cells. The authors should include this information in the review, considering the concentration of nanoparticles and their surface modification. Moreover, they should discuss the issue of biocompatibility of both materials, taking into account their surface properties.
Response 4: As requested, this information has been added.
Comment 5. The issues of protein-corona formation on the surfaces of both metal oxides should be discussed in detail, especially in comparison to the use of these materials for the delivery of other biologically active materials.
Response 5: As requested, this information has been added.
Comment 6. The term nanomaterial should be defined according to the IUPAC definition (page 5). Fig. 3: The captions should be improved – Liposomes instead of Kiposomes.
Response 6: The term "nanomaterial" is defined according to the International Union of Pure and Applied Chemistry (IUPAC), and the corresponding reference is provided. The caption for Figure 3 has been corrected and added.
Comment 7. I suggest using the abbreviation TiO2Ps to indicate TiO2 in the form of particles.
Response 7: We use the abbreviation NPs for the term nanoparticles.
Comment 8. Reactive oxygen species are described with their forms on page 10, although they are discussed earlier. I suggest defining reactive oxygen species in the introduction.
Response 8: According to your suggestion, this request was made.
Comment 9. What does “chemically pure” mean? (page 4)
Response 9: The term "chemically pure" refers to a PS that is a chemically pure compound. The phrase was complete.
Comment 10. The examples of application of TiO2 and ZnO in photodynamic therapy should be discussed in detail. I suggest adding relevant literature reports describing these issues.
I think that more details related to the synthesis and preparation of both types of metal oxides should be included. The authors should also discuss how the preparation procedures and purification methods of these particles influence their biological activity.
Response 10: Thank you for your suggestion. Both topics are important; however, we believe that the application of the two oxides (TiOâ‚‚ and ZnO) in photodynamic therapy (PDT) is more significant. This information has been added to the manuscript. In order to include more details about the synthesis and preparation of metal NPs, we include some characteristics of the formulations actually used, and particularities of the components in the section 4.1, for TiO2 (highlighted in yellow)
In the case of ZnO, the issue was addresses at section 4.2 (highlighted in yellow), in addition some details about the efficacy of formulations of NPs are addressed in other sections.
So that we consider that adding a new section about the subject is some what far from the main objective of the review and may result redundant. We hope this approach fulfill your kind request.
Comment 11. Summarizing, I suggest improving this review to increase its quality.
Response 11: Your comments have certainly helped us improve the quality of the manuscript. Thank you for your observations.

Reviewer 3 Report
Comments and Suggestions for Authors
In this review, the author provides an overview of photodynamic therapy (PDT) using TiO2 and ZnO nanoparticles as photosensitizers in GBM diseases. The discussion on PDT emphasized on its mechanism, advantages, and the enhancement through nanomedicine-based photosensitizers. But for current form, I don’t think this manuscript reaches to the standard for publication, below which several critical issues are outlined:
1) In Abstract, it lacks the clarity and impact; the author should re-write this part with highlight on the rationale and detail outline for exploring PDT as a promising alternative therapy for GBM.
2) Please revise the section numbering for consistency. There are two sections labeled as ‘1. Introduction’ and ‘1. Glioma’.
3) In Part 4. Nanotechnology in PDT, the author provides several paragraphs, like ‘the Materials and Methods......, and GenAI, which appears incorrectly placed or added. Also, the sentences beginning from line 289 to 291 are unreasonable with no relevance to PDT therapy.
4) More insights into the current challenges and future prospects of PDT in GBM treatment should be provided in a detailed way to strengthen the discussion.
Author Response
Reviwer 3
In this review, the author provides an overview of photodynamic therapy (PDT) using TiO2 and ZnO nanoparticles as photosensitizers in GBM diseases. The discussion on PDT emphasized on its mechanism, advantages, and the enhancement through nanomedicine-based photosensitizers. But for current form, I don’t think this manuscript reaches to the standard for publication, below which several critical issues are outlined:
We are grateful for the insightful comments provided by the reviewer, which have contributed to enhancing the quality of the manuscript.
Comment 1. In Abstract, it lacks the clarity and impact; the author should re-write this part with highlight on the rationale and detail outline for exploring PDT as a promising alternative therapy for GBM.
Response 1: The manuscript summary has undergone a restructuring to emphasise photodynamic therapy (PDT) as an alternative treatment for GB.
Comment 2. Please revise the section numbering for consistency. There are two sections labeled as ‘1. Introduction’ and ‘1. Glioma’.
Response 2: As recommended by the reviewer, the numbering of each section in the manuscript has been reviewed and corrected.
Comment 3. In Part 4. Nanotechnology in PDT, the author provides several paragraphs, like ‘the Materials and Methods......, and GenAI, which appears incorrectly placed or added. Also, the sentences beginning from line 289 to 291 are unreasonable with no relevance to PDT therapy.
Response 3: The paragraphs have been removed.
Comment 4. More insights into the current challenges and future prospects of PDT in GBM treatment should be provided in a detailed way to strengthen the discussion.
Response 4: I would like to express my sincere gratitude for your insightful commentary. The present study has expanded the information regarding the use of TiO2NPs and ZnONPs as photosensitizers for GB.

Round 2
Reviewer 2 Report
Comments and Suggestions for Authors
In my opinion, the manuscript titled "Treatment of glioma by photodynamic therapy using TiOâ‚‚ and ZnO nanoparticles as photosensitizers" has been significantly improved by the authors. After carefully reviewing the revised version, I am confident that the authors have adequately addressed all my previous comments and concerns. The manuscript has been improved in both its scientific content and technical presentation. I am confident that this study will make a valuable and significant contribution to our understanding of the role of TiOâ‚‚ and ZnO nanoparticles as photosensitizers in photodynamic therapy, with potential implications for both biology and industry. Therefore, I recommend publication of the manuscript.
Author Response
Dear Reviewer
We are immensely grateful for your feedback. It is important to acknowledge the significant contribution of the observations made by the Reviewer, which have led to substantial enhancements in the quality of the manuscript.
Best Regards
Reviewer 3 Report
Comments and Suggestions for Authors
Accepted!
Author Response

(The authors gave the same response as above.)
